# OpenReview forum: "PETRA: Parallel End-to-end Training with Reversible Architectures"
_NeurIPS.cc/2024/Conference — Submitted to NeurIPS 2024_

### Official Review · Reviewer_YFuR · 2024-07-12

**Soundness:** 1
**Presentation:** 3
**Contribution:** 2
**Rating:** 3
**Confidence:** 4

**Summary:**

In pipelined model training, one important issue is to reduce the bubble sizes.
One stream of work is to use the staleness, where the weight discrepancy is mitigated using stashed weights.
This work tries to reduce the overhead of storing weights with reversible architectures.
Using the non-stashed updated weights, but with restored inputs to each stage,
approximated gradients are obtained and parallel training is performed.
This leads to less memory usage on training at the cost of increased communication.
Training results on resnet variants seem to maintain accuracy.

**Strengths:**

- This is a nice adaptation of reversible architectures to pipelined training. If it works well, there is a potential for becoming a new popular pipelined training method.

- The idea of using reconstructed input instead of stored weights seem to be novel.

**Weaknesses:**

- Insufficient experiment size: Only compared on three different sizes of resnet. This is far from sufficient, especially with the largest model being resnet50.

- No comparison on speedup: speedup on the training time is crucial, but the "memory benefits and training time" section does not disclose any data. Since the proposed scheme has larger communication, it is crucial to report the number.

- Classification accuracy drop: The final accuracy drops on all three datasets for resnet50. 0.6%p and 0.7%p are huge drops for those models. Given that this is the largest model among the tested ones, it draws a significant concern on whether this technique would work for larger models such as resnet152 or ViTs.

- There is no analysis or proof on why the proposed scheme would work. Why it is a good approximation, or why it is going to converge, etc.

**Questions:**

The questions essentially come from the weaknesses.

1. The training time seems to be missing. Could the authors fill it in?

2. How would it work for larger models?

3. Is there a guarantee or an analysis on the convergence or the approximation error?

---

> ### Author Rebuttal · Authors · 2024-08-06
>
> We thank the reviewer for acknowledging the quality and novelty of our algorithm. We respectfully address the weaknesses reported, which imply additional engineering work beyond the scope of this paper:
>
> 1. **Wider Variety of Tasks and Architectures**: While we agree that it would be beneficial to test our model on a wider variety of tasks and architectures, our results are comparable with the best references we are aware of [1].
>
> 2. **Training Time Mention**: We will remove the mention of "training time" as it is already discussed in Table 1. Since our algorithms were executed in a simulated environment, wall-clock training time cannot be used to estimate speedups accurately.
>
> 3. **Drop in Accuracy**: We agree that the drop in accuracy (not observable on ImageNet32) is undesirable. This is likely due to the need to adapt more hyperparameters of our method. The ResNet-50 takes much longer to run with our current experiments, preventing us from conducting larger experiments for now.
>
> 4. **Convergence Analysis**: Some analysis of convergence can be found in [1] for non-reversible architectures. The adaptation to a reversible architecture is straightforward; however, given the non-trivial nature of such proofs, we prefer to avoid mentioning them.
>
> We now answer each of the reviewer’s questions:
>
> 1. **Simulator Evaluation**: We used a simulator to evaluate the stability of our optimization procedure. The expected training time with an effective parallel implementation is presented in Table 1, and [1] demonstrates speedup in a less stale but non-reversible setting. However, more effort is needed in our implementation, as mentioned in our general comment.
>
> 2. **Larger Scale**: The question is relevant since training on the ImageNet dataset was less prone to instability than CIFAR-10, indicating that our method may behave favorably on more challenging benchmarks. Our academic resources do not allow us to benchmark significantly larger models within the rebuttal time frame.
>
> 3. **Convergence Analysis for Reversible Architectures**: See our answer to weakness 4.
>
> \[1\] [On the Acceleration of Deep Learning Model Parallelism With Staleness](https://openaccess.thecvf.com/content_CVPR_2020/papers/Xu_On_the_Acceleration_of_Deep_Learning_Model_Parallelism_With_Staleness_CVPR_2020_paper.pdf)

---

> > ### Comment · Reviewer_YFuR · 2024-08-12
> >
> > Thanks for the rebuttal.
> > However, I believe simulated speedup is alone is insufficient. The algorithm itself has a meaning, but the demonstrated proof and the implementation take a significant part of the contribution. Regarding the hyperparameter tuning, the tuned parameter is also an important part of research. It's important to provide a tuned hyperparameter, and sometime too many hyperparameters to tune is a serious weakness. Therefore, I will maintain my rating.

---

> > > ### Author Response · Authors · 2024-08-14
> > >
> > > We sincerely thank the reviewer for their time during this rebuttal period.
> > >
> > > We tried our best to emphasize the crucial hyperparameters, notably the need for a linear warm-up phase, which has a significant impact on convergence and final model performance.
> > >
> > > Also, achieving practical speedups in a real distributed environment presents distinct challenges from those addressed by our simulator, which was designed to study the convergence of various model parallel approaches; we believe these two tasks require separate, focused investigations. Nevertheless, we fully respect the reviewer's opinion on this issue and thank him for their honest feedback.

---

### Official Review · Reviewer_NpfE · 2024-07-12

**Soundness:** 3
**Presentation:** 4
**Contribution:** 2
**Rating:** 3
**Confidence:** 5

**Summary:**

This paper proposes a method that combines reversible neural networks and parallel distributed training to enable learning with minimal memory usage, while incurring only slight communication and computation overhead. In this approach, the need for storing intermediate activations in traditional backpropagation is eliminated, thus reducing memory constraints and allowing for higher parallelism on the same device. This new method facilitates efficient learning by providing an innovative solution.

**Strengths:**

The problem setup involving reversible architecture and distributed parallel training is intriguing. High memory consumption is a critical issue in learning, and reversible architecture has been proposed to address this problem. It is anticipated that these advantages can be similarly applied to distributed parallel training. Additionally, the paper is very well-written, making the ideas easy to understand. The figures and tables were also judged to be of high quality and well-prepared.

**Weaknesses:**

The main drawback of this paper is the insufficient experimentation. Although using reversible architecture in distributed training is a novel concept, it appears to be merely a combination of existing ideas. For this paper to have a significant impact, it must demonstrate the advantages and benefits of the proposed idea in an actual distributed learning environment. However, the experiments were conducted using only a single A100 GPU, and there is no demonstration of the performance improvements or limitations of the proposed idea in a real distributed environment. The values presented in the tables do not clearly differentiate from what can be achieved with existing reversible architectures. To improve the completeness of this paper, it is essential to analyze scenarios that necessitate the use of multiple GPUs, such as video applications, large-resolution diffusion, and large language models. The current data fails to effectively explain the benefits of the proposed idea.

**Questions:**

1. Additional data is needed to determine the advantages in an actual distributed environment and to assess how much the increase in communication and computation affects speed.

2. Additional experiments are essential to understand how the required minimum number of GPUs changes for tasks that require a lot of memory, or how the tasks that can be run on the same cluster are affected.

3. As mentioned in the conclusion, experiments with large language models are essential.

4. Detailed experiments are also needed to see how these experiments impact accuracy and to analyze the effects on complex tasks and hyperscale models.

5. Where does the accuracy difference between RevNet backprop and PETRA in Table 2 arise? Besides the subtle errors that can occur in precision, are there any other potential sources of error?

**Limitations:**

Not relevant.

---

> ### Author Rebuttal · Authors · 2024-08-06
>
> We thank the reviewer for acknowledging the quality of the ideas presented in this paper. We would like to address the concerns:
>
> 1. Our paper serves as a proof of concept for a new algorithm, and our resources (engineers, clusters) currently only allow us to use a simulated environment. Several works have demonstrated that the use of stale gradients can be implemented efficiently (see [1,2]). However, our focus is to demonstrate that our method attains high accuracy with a larger number of stages while offering a drastic reduction in memory footprint. Although theoretical, Table 1 details the issues (buffers, communications, compute time per batch) concisely and precisely.
>
> 2. We agree that comparing the minimum number of GPUs required for a given task with different optimization methods, or how the maximum size of a model that can fit on a given cluster changes for different optimization methods, is crucial to emphasize the practical advantages of our method against available alternatives. Despite our implementation effectively simulating PETRA and alternative optimization methods within the same code base, a distributed implementation is necessary for such comparisons, which is still under development.
>
> 3. Our method used the maximum number of stages for our models, i.e., $10$ for ResNet-18 and $18$ for ResNet-50, unlike the closest work to ours, which only goes up to $K=3$ on ImageNet (see [1]). Extending our work to other downstream tasks, such as large language model (LLM) training, is interesting but beyond the scope of this work.
>
> 4. As mentioned in response to the first and second weaknesses, our current resources prevent us from experimenting at a larger scale. However, such experiments are under active development, and we emphasize that we are not aware of other works allowing such an extensive split of $18$ stages while keeping good performances as we showed here.
>
> 5. Between the reversible backpropagation of the RevNet and PETRA, there are two differences. The first is the use of stale gradients to update the parameters, which is key to the linear speed-up of such approaches. The second difference is the removal of the parameter buffers in the stages. This results in PETRA approximating the backpropagation computation with the updated parameters, which differ from the ones in the forward pass. However, we found this approximation to have a limited impact. Thus, minor fluctuations are expected due to the use of stale and approximated gradients, but they are surprisingly low. Additionally, we had to fully reimplement autograd to make the gradient estimation process feasible, so errors like round-off are common and expected.
>
> \[1\] [On the Acceleration of Deep Learning Model Parallelism With Staleness](https://openaccess.thecvf.com/content_CVPR_2020/papers/Xu_On_the_Acceleration_of_Deep_Learning_Model_Parallelism_With_Staleness_CVPR_2020_paper.pdf)
>
> \[2\] [PipeDream: Fast and Efficient Pipeline Parallel DNN Training](https://arxiv.org/pdf/1806.03377)

---

> > ### Comment · Reviewer_NpfE · 2024-08-11
> > **Response to the rebuttal**
> >
> > Thank you for your sincere effort. I agree that the proposed idea is interesting. However, I believe it is crucial to provide more practical and realistic results to demonstrate the benefits of the proposed approach. The current version relies heavily on simulation and estimation, which is not sufficient for papers that are closely related to systematic applications like this. Therefore, I will maintain my decision.

---

> > > ### Author Response · Authors · 2024-08-14
> > >
> > > We sincerely thank the reviewer for their time during this rebuttal period.
> > >
> > > Achieving practical speedups in a real distributed environment presents distinct challenges from those addressed by our simulator, which was designed to study the convergence of various model parallel approaches; we believe these two tasks require separate, focused investigations. Nevertheless, we fully respect the reviewer's opinion on this issue and thank him for their honest feedback.

---

### Official Review · Reviewer_E8Gz · 2024-07-13

**Soundness:** 3
**Presentation:** 3
**Contribution:** 3
**Rating:** 5
**Confidence:** 3

**Summary:**

In this paper, the author proposes a new alternative algorithm (Parallel End-to-End Training with Reversible Architectures) for regular backpropagation, which significantly enhance the parallelization with a limited overhead compared to regular backpropagation and other alternatives to end-to-end training.
Specifically, the network is split into several stages (one layer or a set of layers) distributed across distinct devices, one batch data is split into several mini-batch data. The first device sequentially accesses the mini-batch data and pass them forward to the next stage until the final stage is reached. The backpropagation is initialized from the final stage to the first stage. It enables a significant parallelization of forward and backward computations across multiple devices.

**Strengths:**

* This paper is well-organized and easy to follow.

* The background information is very rich and makes it easy for someone who is not familiar with this field to understand the relevant techniques including the technique proposed by this paper.

* The figures about the core technique proposed by authors are very clear, which can help readers understand the technique at a glance.

* The paper evaluates the proposed techniques on multiple datasets and networks.

**Weaknesses:**

* From the comparison between the proposed method and other techniques from related work, it showcases that the proposed method does not have an overall crushing lead. There exists the method which can achieve higher speed and less time than proposed method with storage increased.

* The low or even zero storage on proposed method is mainly due to reversible architectures. Maybe authors can extend proposed parallel training method to some non-reversible architectures (need memory storage for intermediate activations), then compare with other SOTA methods.

* It would be great if authors use more distributed devices to get more stages from a network, in this case, the performance of the proposed method is likely to be deeply explored. Because the proposed technique is aimed to deployed on the distributed devices.

**Questions:**

Please refer to the weakness section.

**Limitations:**

Yes.

---

> ### Author Rebuttal · Authors · 2024-08-06
>
> First, we sincerely thank the reviewer for their positive assessment and acknowledgment of the clarity and novelty of our method. We would like to address the reported weaknesses:
>
> 1. We believe the reviewer is referring to Table 1. We would like to reformulate our discussion from the end of Section 3.3: Except for PETRA, each method in the table has a dependency on $J$, the depth of the network. For example, backpropagation for reversible architectures has a mean time per batch that depends on $J$, meaning it cannot achieve computational speedup. For delayed gradient methods, the size of the buffers depends on $J$, leading to strong memory constraints. As $J$ grows, only PETRA maintains a constant use of memory resources.
>
> 2. To our knowledge, the best comparison point is [1], which does not implement any invertible mechanism. Their splits into $K$ stages are small ($K=3$), whereas we use $K=18$. Table 4 of this paper reports top-1 accuracies of 68.9% for ResNet-18 and 74.9% for ResNet-50 using a split of 3. In comparison, using splits of $K=10$ and $K=18$ respectively, we report top-1 accuracies of 71.0% and 74.8%, which is significantly better. This paper was published at CVPR, a top-tier conference, with an optimized implementation that is not available online. Therefore, we could not base our work on it, even though they report a computational advantage with only a split of $K=3$.
>
> 3. We agree that deploying our code in a real distributed environment to benchmark its efficiency would be the ideal proof of concept. However, our academic resources (engineers, clusters) only allowed us to use a simulated environment. Nevertheless, this environment enabled us to accurately simulate many model parallel approaches and demonstrate the convergence of PETRA, even for ImageNet with many stages ($18$), accomplishing the goal of this paper.
>
> \[1\] [On the Acceleration of Deep Learning Model Parallelism With Staleness](https://openaccess.thecvf.com/content_CVPR_2020/papers/Xu_On_the_Acceleration_of_Deep_Learning_Model_Parallelism_With_Staleness_CVPR_2020_paper.pdf)

---

### Official Review · Reviewer_uNwa · 2024-07-13

**Soundness:** 2
**Presentation:** 3
**Contribution:** 2
**Rating:** 3
**Confidence:** 3

**Summary:**

The authors propose fusing delayed gradient pipeline parallelism with reversible models in order to capture the benefits of the former while mitigating the drawbacks with the latter.

**Strengths:**

- The paper sets up a pretty compelling combination of ideas. This is a great example of a paper that clearly understands the strengths and weaknesses of two disparate techniques and fits them together like puzzle pieces.

- The paper is clear and methodical in laying out the motivation for the approach. By the time the method is introduced, its seems like the natural and obvious choice. This is good writing.

- The concept is solid. I really *want* to like this idea, since it seems to fit together so well.

**Weaknesses:**

- While the idea is presented fairly clearly, a lot of the analysis is estimates (S4.2) and generalizations (Tab 1). It's fine for motivating the idea, but not really good enough for proving it works as projected. I'm left wondering how much of this method will actually translate to a scaled-up implementation. (No question that it *was* implemented, but a pipeline-parallel model that doesn't actually pipeline across devices is...not particularly compelling.)

- The paper is a fusion of two ideas, designed to capture the computational performance benefits of pipeline parallelism while using reversible models to mitigate memory scaling. Some estimated results of memory footprint are presented in Table 3. No measured results are presented related to parallelism (timing, utilization, etc.). From this paper, it is not possible to determine whether it has succeeded. This is confused further by the section 4.2: "Memory benefits and training time" which does not discuss training time at all. The lack of computational results is fairly damning.

**Questions:**

Q1. Most distributed parallel solutions emphasize minimizing interconnect traffic, as it often bottlenecks overall runtime. Table 1 suggests PETRA would increase inter-pipeline-stage traffic by 2-4x (forward/backward). This would suggest that a scaled-up implementation might struggle under increased communication. As the presumed goal of PETRA is to scale up (L64, L115), how do the authors mitigate the increased communication?

(minor) The last column in Table 1 is not described in text. Is this just highlighting that these two are pipelined?

**Limitations:**

As described in weaknesses. Limitations, like computational performance details, are not well described.

---

> ### Author Rebuttal · Authors · 2024-08-06
>
> First, we sincerely thank the reviewer for the very positive feedback. It's truly appreciated.
>
> We'd like to address the weaknesses you reported:
>
> 1. We agree that developing a scalable and distributed implementation of the algorithm would be the ultimate proof of concept. However, our primary goal is to introduce a novel training algorithm that could lead to a significant breakthrough. We have achieved this goal by proposing a fully functional simulator of our method and other model parallel approaches (DSP [1], PipeDream [2]), which allowed us to empirically confirm the success of our method on a large-scale dataset for a high number of stages (18). While many questions remain open and a functional distributed implementation is essential for engineering purposes, we believe that NeurIPS is the right venue to explore new, promising ideas.
>
> 2. We will remove the mention of “training time” in the title of the paragraph at line 256, as this was a mistake since it is already discussed in Table 1. The memory savings in Section 4.2 are reported within the context of our simulator, which measures the impact of maintaining parameter or activation buffers. Table 1 effectively explains the benefits of our method compared to state-of-the-art approaches, including pipelining, which necessarily involves some additional buffers to handle microbatches. Our main objective was to remove the dependency on $J$, which is a problematic scaling law in both time and space categories of the Table. Practical measurements are highly code-dependent, and we believe that such low-level optimization and benchmarking are out of the scope of this paper.
>
> Regarding the question about the impact of interconnect traffic, we acknowledge that it can be a serious issue in distributed training. However, since our increase factor for communication is fixed (i.e., $2$ and $4$ for forward and backward communication, respectively), we do not believe it is a fundamental concern for the scaling potential of PETRA. In model parallelism, the communications between workers typically contain activation tensors, which have sizes proportional to the *width* of the model. If PETRA is dominated by communications during training on a given cluster for a reversible architecture, it means that any other model-parallel method would struggle to scale the *width* of the non-reversible counterpart beyond a factor of $2$ or $4$. A similar argument can be made regarding the mini-batch size, which also scales with the activations. Thus, while our increased communications are important to understand for making the best use of a given compute configuration, their increase is only constant and does not overshadow PETRA's main advantage: a linear speedup for a constant memory cost, allowing significant depth scaling due to substantial memory reduction.
>
> \[1\] [On the Acceleration of Deep Learning Model Parallelism With Staleness](https://openaccess.thecvf.com/content_CVPR_2020/papers/Xu_On_the_Acceleration_of_Deep_Learning_Model_Parallelism_With_Staleness_CVPR_2020_paper.pdf)
>
> \[2\] [PipeDream: Fast and Efficient Pipeline Parallel DNN Training](https://arxiv.org/pdf/1806.03377)

---

> > ### Comment · Reviewer_uNwa · 2024-08-11
> >
> > 1. The paper indirectly claims computational benefits but does not measure computational performance. Pipeline parallelism is used to improve computational performance. If you are claiming that you're combining the benefits of reversible models with pipeline parallelism, then the claim is that you're keeping the computational benefits of pipeline parallelism. There's no empirical evidence provided. I agree with the authors that it's not be necessary to implement production-quality, cluster-scale version of this, but it's necessary to show that the technique doesn't decimate the computational benefits of pipeline performance. The current paper is not sufficient in that respect.
> >
> > 2. On the contrary, I would suggest you keep training time in that section, but fill it out with training time data measured from a real implementation across multiple devices compared with other pipeline parallel frameworks, since that's the intended use case and your competition. I strongly disagree that practical measurements are "low-level optimizations" or out of scope.
> >
> > I would like to point out to the authors that the pipeline parallelism papers they cite in this work do indeed provide empirically-measured timing numbers for their approaches. This is the accepted bar to meet, and I think it is reasonable to expect that here.

---

> > > ### Author Response · Authors · 2024-08-14
> > >
> > > We sincerely thank the reviewer for their time during this rebuttal period.
> > >
> > > We claim a computational advantage because our algorithm can be parallelized since the workers operate independently of each other. The PipeDream paper does provide empirical speedups, and it is proof to us that such model parallel approaches are relevant for making the best use of hardware. Achieving practical speedups in a real distributed environment presents distinct challenges from those addressed by our simulator, which was designed to study the convergence of various model parallel approaches; we believe these two tasks require separate, focused investigations.
> > >
> > > Also, since our gradient estimator is mathematically novel, we need to empirically validate our new learning dynamics on standard benchmarks to justify going beyond this point. We do this with a larger number of stages than is usually done in related publications to investigate the resilience of the algorithm with respect to staleness, since our goal was to show that we can scale the number of workers without degrading performance, which is non-trivial with delayed gradient approaches.  We believe that such a simulation was necessary before focusing our efforts on an implementation capable of exploiting distributed environments in practice.
> > >
> > > Nevertheless, we fully respect the reviewer's opinion on this issue and thank him for their honest feedback to guide our investigation and improve our paper.

---

### Author Rebuttal · Authors · 2024-08-06

We appreciate that every reviewer acknowledged the refreshing aspect and elegance of our method. Several reviewers noted the lack of empirical data on the efficiency of our method in a distributed environment at scale. We emphasize that this work represents the beginning of a promising line of research, which we are progressively testing on increasingly complex benchmarks within a simulator.

Implementing our approach on a large scale is highly challenging and requires significant engineering effort. To reach its full speedup potential, our method must effectively overlap computation and communication, a non-trivial optimization task. Additionally, ensuring compatibility with traditional pipelining approaches within the same code base is crucial for accurate benchmarking of LLM training efficiency. We consider this an independent project of considerable scope, requiring **significant funding**, to be pursued once the approach has been thoroughly validated in a realistic simulated environment, which is already highly resource-intensive for an academic setting. We can accurately simulate backpropagation, PETRA, and approaches like PipeDream [2] and DSP [1], but without leveraging their distributed advantages yet. Since our focus is not specifically on LLMs, we do not overstate our implementation's current capability to train them at scale.

It is important to note that pipelining approaches like GPipe [3] do not actually change the **training algorithm**, but mainly require **challenging implementations** to improve training speed. In contrast, approaches that allow delayed gradients like [1, 2] decouple the layer computations and remove worker idleness, but this comes at the cost of modifying the **training dynamic** and incurring a **quadratic total memory overhead** for activations and parameters. We thus believe that PETRA is a highly relevant alternative to current training techniques, maintaining the **linear speedup** of such approaches while keeping a **constant memory cost**.

The closest reference to our work, which minimizes the use of buffers, is [1]. However, their experiments were limited to a **small number of stages** (e.g., 3 stages for ImageNet) compared to up to **18 stages** in our case without performance degradation. Their work has had limited adoption as they did not release the source code for their experiments. Our code base, while not yet achieving speedup, **covers many model parallel approaches** and is under active development to explore the feasible limits of model parallel training techniques.
Finally, we firmly believe that NeurIPS is the ideal venue for presenting and proposing novel algorithmic directions that significantly depart from existing approaches. The absence of a large-scale, state-of-the-art implementation across multiple modalities (e.g., LLM) and without the use of massive clusters should not detract from the significance and novelty of our work. To our knowledge, our method is the first to achieve **a true decoupling of the forward and backward procedures** since we do not maintain any buffer between executions. We believe that our innovative approach stands on its own merit.


\[1\] [On the Acceleration of Deep Learning Model Parallelism With Staleness](https://openaccess.thecvf.com/content_CVPR_2020/papers/Xu_On_the_Acceleration_of_Deep_Learning_Model_Parallelism_With_Staleness_CVPR_2020_paper.pdf)

\[2\] [PipeDream: Fast and Efficient Pipeline Parallel DNN Training](https://arxiv.org/pdf/1806.03377)

\[3\] [GPipe: Efficient Training of Giant Neural Networks using Pipeline Parallelism](https://arxiv.org/abs/1811.06965)

---

### Decision · Program_Chairs · 2024-09-25

**Decision:**

Reject

**Comment:**

Thanks for your submission to NeurIPS.

This paper seems to have a lot of potential.  The reviewers in general really liked the idea, liked the clarity of presentation, and appreciated the concepts involved.  Unfortunately, even after the discussion period, the reviewers were generally in agreement that, in its current form, the paper is not yet ready for publication.  Each of the reviewers had comments related largely to the empirical study of the paper, where it seems that one needs much further validation of the proposed approach.

I suspect that this paper could be very successful indeed, with some additional work and more experimental validation.  But in its current form, there is simply not enough support amongst the reviewers to warrant acceptance at this time.